# A Review of Non-Chemical Weed Control Practices in Christmas Tree Production

**Debalina Saha [1,\*], Bert M. Cregg [2] and Manjot Kaur Sidhu [1]**

[1] Department of Horticulture, Michigan State University, 1066 Bogue Street, East Lansing, MI 48824, USA; sidhuman@msu.edu

[2] Departments of Horticulture and Forestry, Michigan State University, 1066 Bogue Street, East Lansing, MI 48824, USA; cregg@msu.edu

\* Correspondence: sahadeb2@msu.edu; Tel.: +1-517-353-0338

**Abstract:** Weeds interfere with Christmas tree growth at any time and at any stage of production. Growers mostly rely on mechanical mowing and applications of herbicides for weed control in their fields. However, herbicides can be phytotoxic to non-target plants, can cause environment-related issues, and their repeated application can even cause herbicide-resistant weeds. The main objective of this manuscript is to provide a review of non-chemical weed control strategies in Christmas tree production and identify areas where current practices could potentially be improved or in which further research is required. Preventing the introduction of weed seeds requires controlling weeds along farm roads, maintaining clean equipment, and eliminating new weeds before they start seeding. Mowing helps to reduce the number of seeds produced by the weeds and can significantly reduce competition with trees. Shropshire sheep are well suited for grazing Christmas tree plantations as they prefer grazing on grasses and weeds rather than on coniferous trees. Weeds can also be controlled around Christmas trees by mulching. Organic mulch can improve soil moisture, maintain soil temperatures, enhance root establishment and transplant survival, and improve plant establishment and overall growth. Incorporating cover crops into Christmas tree plantations may improve tree growth, quality, and soil fertility and can supplement conventional nitrogen fertilizers. However, if cover crops are not properly managed, they can be highly competitive with the trees. Flaming can cause suppression of many annual weed species but is less effective on larger weeds and needs to be applied with caution. Several insects have been used as biological agents to control selective weed species. However, further research is required to focus on several potential biological agents, different types and depths of mulches, on cover crops types and their competition with different species of Christmas trees and their effects on seedling survival and growth.

**Keywords:** animal grazing; biological weed control; cover crops; mechanical weed control; organic mulching; weed competition

## 1. Introduction

According to the author of Reference [1], there are about 141,640 hectares in production for growing Christmas trees on about 15,000 farms across the United States. More than 100,000 people are employed full- or part-time in this industry with an average of $250 million in sales per farm per year [2]. Michigan alone supplies about two million fresh trees with an annual net value of $30–40 million to the national market each year, which are grown on 14,973 hectares of land [3]. To improve the production and profit margins of this industry, Christmas trees must be established correctly and properly maintained.

Weeds in Christmas tree plantations can reduce leaching of nutrients, improve microclimates, reduce the risk of wind and water erosion, and can improve biodiversity [4]. However, it is also important to strike a balance between protecting soil and water and reducing weed competition with trees [5]. Weed control is an important aspect to consider for successful Christmas tree production from both an aesthetic and biological perspective. To produce high-quality, marketable Christmas trees, a good weed management plan is essential [5]. Weeds interfere with tree growth at any time and at any stage of production [6] as they can compete for nutrients, water, space, light, oxygen and can even harbor pests and pathogens. Effective weed control is extremely important for initial conifer seedling survival and in the first three years after transplanting (during the establishment period) in the field [6–10]. The establishment phase of Christmas trees is critical as weed competition, particularly for water [11], can lead to suppression of tree growth and can even cause death of the trees. In the second and third year of the establishment phase, the rate of tree growth is directly related to the amount of weed competition [6]. In sandy soils, weeds can deplete the limited available moisture and ultimately the trees can succumb to drought stress. Christmas tree seedlings in plantations can be over-topped and shaded by weeds. This can result in reduced photosynthate production, which can reduce leaf area development and subsequent growth [12]. In later stages of rotation, when trees are larger, weeds interfere with production practices such as pruning and spraying pesticides [6] and can even shade lower branches of trees [13]. Broadleaves weeds such as field bindweed (*Convolvulus arvensis* L.), horseweed [*Erigeron canadensis* (L.) Cronquist], common ragweed (*Ambrosia artemisiifolia* L.) and giant ragweed (*Ambrosia trifida* L.), wild carrot (*Daucus carota* L.), hoary alyssum [*Berteroa incana* (L.) DC.], and hairy vetch (*Vicia villosa* Roth)—as well as seedheads of grasses such as giant foxtail (*Setaria faberi* Herrm.), witchgrass (*Panicum capillare* L.), large crabgrass [*Digitaria sanguinalis* (L.) Scop], and fall panicum (*Panicum dichotoliflorum Michx*) can grow into the tree branches and can be difficult to remove. Vining weeds such as field bindweed, Virginia creeper [*Parthenocissus quinquefolia* (L.) Planch.], poison ivy [*Toxicodendron radicans* (L.) Kuntze] and wild grape (*Vitis* spp. L.) can grow into Christmas trees. In these cases, they cannot be treated with herbicides without risk of injury to the trees [6]. The lower branches of Christmas tree can become abraded by weeds which can cause the browning and dropping of lower needles, and ragged crown growth [14].

Christmas tree growers mostly rely on mechanical mowing and applications of chemical herbicides to control the weeds in their production fields. However, repeated applications of the same herbicides have resulted in development of herbicide-resistant weed species. In Michigan there are recent reports on common ragweed resistance to clopyralid (Stinger), a synthetic auxin herbicide, by some Michigan Christmas tree growers, especially in Montcalm County [15]. Many postemergence herbicides can even cause severe phytotoxic injuries to different species of Christmas trees including stunted growth, burning and dropping of needles, chlorosis, and even complete death of the tree. During the establishment phase in particular, young Christmas trees are sensitive to these chemical herbicides. In addition, there can be negative environmental issues such as herbicide leaching, drift, and run-off. Due to the issues surrounding herbicide use and increasing concern among some consumers related to pesticide applications, many growers are interested in strategies to reduce or even eliminate chemical weed control. The purpose of this manuscript is to provide a review on nonchemical weed control strategies in Christmas tree production and identify areas where current practices could potentially be improved or in which further research is required.

## 2. Prevention

Prevention of weed seed introduction to the Christmas tree field is the first step to effective non-chemical weed control. Although natural dispersing agents including wind, water, birds, mice, etc. can move weed seeds into fields, weed seeds, rhizomes, parts of roots and stems can be also moved from site to site on equipment or clothing. Preventing the introduction of weed seeds requires controlling weeds along farm roads, maintaining clean equipment and equipment yards, and eliminating new weed

species before they start seeding or become established. If topsoil is introduced to a site, it should be clean and free of roots, rhizomes, seeds, and other weed propagules. Regular scouting for weed species needs to be undertaken by growers in their fields and immediate hand removal is encouraged as later mowing in between the Christmas tree rows can spread the seeds and rhizomes. According to a case study by the authors of Reference [5] there was a recent increase in puncturevine (*Tribulus terrestris* L.) in the Willamette Valley of Oregon, United States due to the seed movement. The fruits of puncturevine stick to tires of tractors and shoes of humans, allowing it to spread very easily over large distances. Once the puncturevine weed is established, it is very difficult to control or eradicate because of its long seed dormancy.

## 3. Mechanical Control

Mechanical control, which includes cultivation and mowing, can help in preventing development of herbicide-tolerant resistant weed species.

### 3.1. Cultivation

Weeds around the bases of the Christmas trees can be controlled by hand or mechanical cultivation [16]. Cultivation can be helpful in controlling small seedlings of many weeds for the first 2 or 3 years after transplanting the Christmas trees. Narrow rototillers, harrows, small disks, and a wide variety of sweeps or other tools designed to cut weeds off just under the soil surface, are some of the cultivation equipment that are currently used in Christmas tree plantation [5]. Cultivation requires more frequent application than other methods for effective control and is not feasible on sloped land due to possible erosion, and can damage tree roots [16]. Moreover, overuse of cultivation can spread perennial weed species such as field bindweed or Canada thistle [*Cirsium arvense* (L.) Scop.] throughout the field, can increase cultivation-resistant species such as common purslane (*Portulaca oleracea* L.), or can damage soil structure in later years [5]. As Christmas trees grow and fill in spaces between rows, it can become more difficult to get cultivation equipment through the plantation and cultivation becomes less effective for controlling weeds.

### 3.2. Mowing

Mowing at the right time (Figure 1) prior to weed seed development helps to reduce the number of seeds produced by the weeds and can significantly reduce the weed competition with the trees.

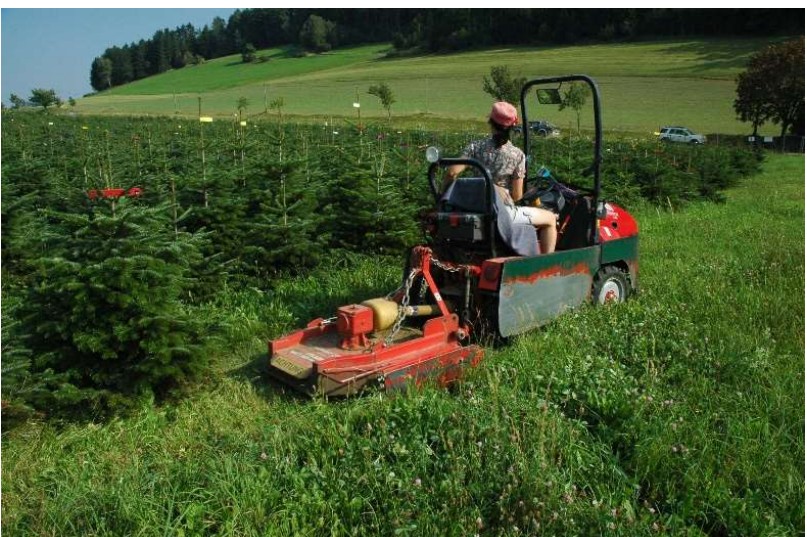

**Figure 1.** Mowing is being performed at the Christmas tree plantation. Mowing at the proper time can reduce weed seed production and thereby can significantly reduce weed competition with Christmas trees.

With large mowing equipment, the potential for damaging the trees increases. Hence the use of deflection shields often becomes necessary to minimize damage to larger trees. Growers need to prevent trunk damage to reduce entry ports for disease and insect organisms [16]. Where weeds immediately around trees and within rows are not controlled by mowing, weed or mechanical trimmers are often used. These weed trimmers can serve dual functions and can be used in the shearing operation by adding of a cutting blade. These trimmers are available in a variety of styles with different power sources, some trimming with a monofilament line, some with metal or plastic blades [16].

## 4. Domestic Animals

Mowing combined with other weed control methods (such as sheep grazing) and at certain times in the growing season or rotation can be effective in controlling weeds [4]. Although grazing animals often can cause weed problems, adjusting grazing timing or intensity or both can sometimes address the problem [17]. Livestock normally do not browse on Christmas trees if other good forage is available, although they can cause damage by brushing against large trees or stepping on small trees [16]. Sheep are less likely to be a problem than cows or horses [16]. According to the author of Reference [18], trials in Denmark and England indicate that Shropshire (or Shrops) sheep are well suited for grazing Christmas tree plantations as they prefer grazing on grasses and other weeds than on coniferous trees (Figure 2). Shropshire sheep became popular in the U.S. in 1920s and 1930s and were originally bred in the U.K. This particular breed is noted for good wool and meat production, having a mild temperament, and being hardy [18].

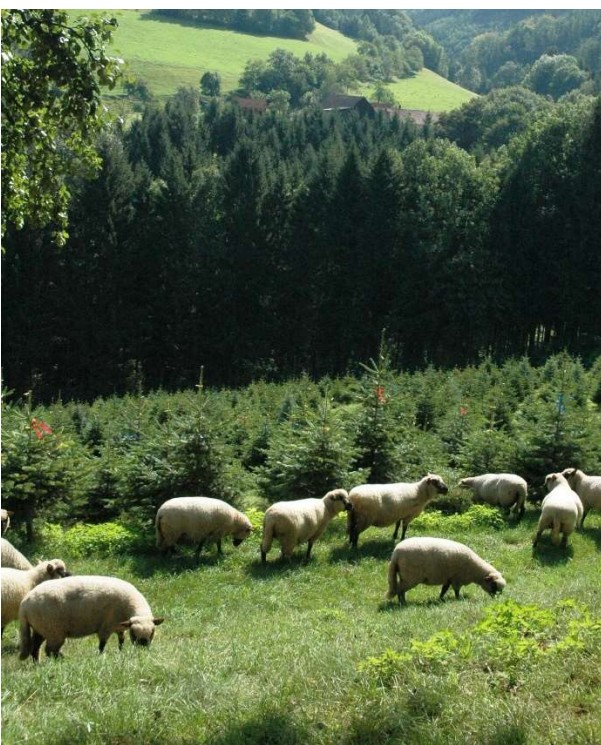

**Figure 2.** Shropshire (or Shrops) sheep grazing in Christmas tree plantation. They prefer grazing on grasses and other weeds than on Christmas trees and thereby help in controlling weeds.

## 5. Mulching

### 5.1. Organic Mulching

Vegetation can be controlled around Christmas trees by one or more varieties of organic mulch which may include grass clippings, nut hulls, wood chips, compost, bark, sawdust, and other organic materials [16].

Organic mulch (Figure 3) can improve soil moisture, reduce soil erosion and compaction, maintain optimal soil temperatures, increase soil nutrition, enhance root establishment, and transplant survival, and improve plant establishment and overall tree growth [19]. Weeds can increase daily evapotranspiration of soil moisture by 25% in summer [20]. In contrast, organic mulches can increase soil moisture content by increasing percolation and retention, reducing evaporation, and reducing weeds [19]. An appropriate mulch layer will significantly reduce the amount of irrigation needed, and in certain cases can eliminate it altogether [21]. Coarse and large particle size organic mulch such as pine bark and pine straw have shown better weed control than fine particle-sized hardwood mulch [22]. Coarse organic mulches can hold water much like a sponge, thereby capturing rainfall and irrigation water for later release and preventing runoff [19]. An early study demonstrated that 1.5 cm of straw mulch reduced water runoff by 43% [23]; mowed sod and bark were likewise found to reduce runoff [24]. In addition to drought stress, mulch can also protect trees from other environmental stresses such as cold injury [25]. Organic mulches help to maintain optimal soil temperatures in that soils can be kept cooler in hot conditions [26–28] and warmer in cold conditions [29]. Temperature modification is extremely important near the soil surface as hot or cold surface soils can kill new transplants that have not had time to generate a large root mass and establish into deeper, more moderate, surrounding soils [19]. Numerous studies have demonstrated that improved water retention and reduced weed growth are correlated with increased root growth and exploration by desirable plants [30,31]. Hence, mulches allow roots of trees to extend and establish far beyond the trunk compared to bare soil and thus become increasingly stabilized [32,33]. Choosing the right type of organic mulch is important in determining how well roots of Christmas trees will expand in the underlying soil. It was observed that root development and density was greatest under organic mulches compared to plastic mulch [30], bare soil [33] or living mulches [34].

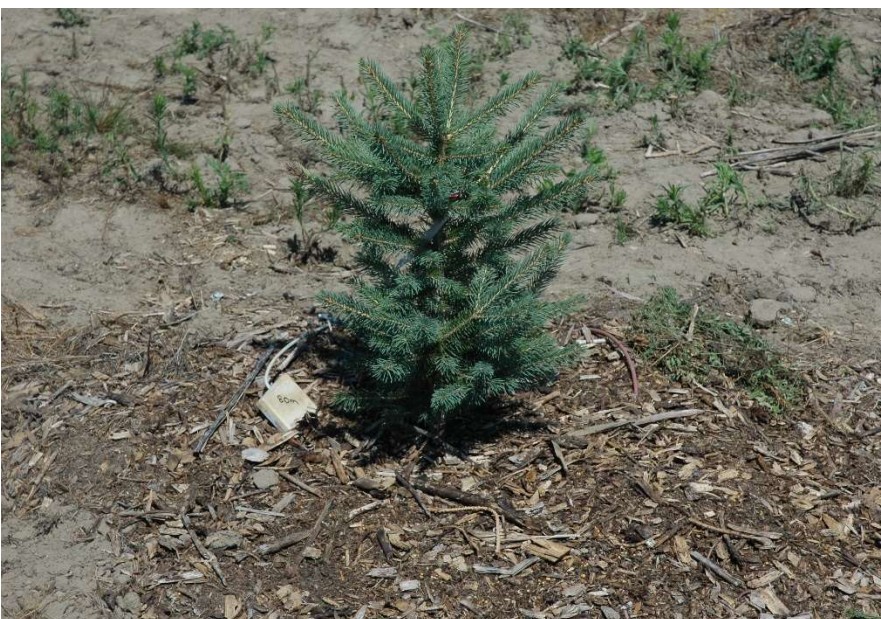

**Figure 3.** Hardwood chips used as organic mulch can prevent weed emergence and growth during the establishment phase of the Christmas trees. This hardwood mulch can also improve soil moisture content, maintain optimal soil temperature, and improve Christmas tree overall growth.

In a study conducted by the authors of Reference [35] on the growth and physiology of newly planted Fraser fir [*Abies fraseri* (Pursh) Poir.] and Colorado blue spruce (*Picea pungens* Engelm.) in response to different types of mulch and irrigation, organic wood chip mulch without any irrigation, improved weed control and resulted in survival and growth that was comparable to trees under irrigation for both species. Organic mulches have an additional advantage of releasing nutrients

as they decay. It is recommended to use old (>6 months) sawdust, bark, or chip mulch so that the initial decay has started, and the time required for early nitrogen demand has already passed [16]. A field experiment was conducted by the authors of Reference [36] in a Christmas tree plantation in eastern Kentucky to examine the effects of four weed-management strategies—sulfometuron methyl herbicide (Oust) at two application rates, organic sawdust mulch, and rubber-tire mulch—on soil nutrients and microbial biomass. The results showed that the Oust and rubber-tire mulch treatments had negative cumulative and re-application effects on soil cations and pH. In contrast, significantly higher microbial biomass occurred in the organic sawdust treatment which was due to increased soil water content under the sawdust mulch and higher total nitrogen and soil organic matter for the cumulative and re-treatment periods. Authors of Reference [36] also identified sawdust as the best of the four weed-control strategies for long-term Christmas tree production as it had positive effects on soil physical, chemical, and biological characteristics, which are important considerations for growers choosing a weed-control strategy for use on land expected to produce Christmas trees for many years.

*5.2. Inorganic Mulching*

Inorganic mulch materials act as physical barriers that limit or exclude weeds and regulate soil temperature and moisture, which aids in seedling establishment and survival. True physical barriers, which do not allow water percolation into the soil but restrict the amount of evaporation away from the soil, include asphalt shingles or plastic sheets of various kinds [16]. Porous inorganic mulch material such as ground rubber does allow rainwater percolation [16]. Solid inorganic surfaces such as concrete [19] and synthetic mulches including asphalt, fabrics and plastics are poor at controlling soil temperatures [37–39]. However, chunky inorganic mulches such as gravel and lava rock are more effective temperature moderators [38,40,41]. Soil temperatures may be raised [37] or lowered [42] by black plastic mulch, depending on how much light is absorbed by the plastic and whether heat is retained or reflected [19]. The authors of Reference [43] conducted a trial on weed control measures in Christmas tree plantations of *Abies nordmanniana* (Steven) Spach and *Abies lasiocarpa* (Hook.) Nutt. in southwestern Norway. Weed control treatments included chemicals, use of black plastic mulch, grass, or clover as ground cover, living mulch and mechanical hoeing. Ground cover with black plastic mulching resulted in the best tree growth and quality. Most of the research on inorganic mulches has focused on black plastic sheet for weed control in Christmas tree production and very little or no research has been conducted with gravel, fabrics, lava rocks and other inorganic mulches.

## 6. Cover Crops and Vegetated Strips

Cover crops can directly compete with weed species and can suppress weed emergence. In addition to weed suppression, cover crops can reduce soil erosion, improve soil physical properties and nutrient content. Incorporating cover crops into Christmas tree plantations may improve tree growth and quality, soil fertility and can be an alternative to commercial nitrogen fertilizers [44]. Cover cropping is widely recognized and promoted as a practical method to enhance soil productivity and environmental quality [39,45]. Groundcover management involves the use of mowed, tilled, or killed cover crops to add organic matter to soil, conserve soil humus, reduce soil erosion, increase soil organic matter levels, and steadily release available nutrients for associated or succeeding Christmas tree uptake as the organic matter breaks down [46,47]. Planting either legume cover crops [e.g., alfalfa (*Medicago sativa* L.) and white clover (*Trifolium repens* L.)] or grasses [e.g., perennial ryegrass (*Lolium perenne* L.)] in the interspaces of Christmas tree can increase plant residue inputs to soils and, therefore, may stimulate soil microbial activity [48,49]. Because cover crops can produce significant amounts of biomass, thrive when repeatedly mowed, fix atmospheric nitrogen through symbiotic association with nitrogen-fixing bacteria, and/or scavenge excess nitrogen left in the soil that would otherwise be lost by leaching, they are considered as ideal candidates for "living mulch" [50,51]. Agricultural management practices that leave plant residues on the soil surface, such as cover cropping and no tillage, often result in higher

concentrations of soluble organic carbon compounds, which may greatly influence soil microbial populations and activities [52,53].

According to the authors of Reference [5], narrow seed drills need to be used or small-seeded cover crops should be broadcast at high rates to establish cover crops after Christmas trees have been planted. Growers need to make sure the seeded area is weed-free prior to seeding. In Christmas tree plantations, hard fescue grasses are common cover crop species. The authors of Reference [44] assessed whether soil fertility, tree survival and growth could be improved by incorporating leguminous and non-leguminous cover crops into Fraser fir production system. Alfalfa, Dutch white clover and perennial ryegrass were grown in a newly established Fraser fir plantation using two cover crop management practices [no banding (NB) or banding (B)] by creating a 61 cm wide bare zone centered on the tree rows and a conventionally-managed system (CONV) was used as a control. Tree seedling survival, growth, photochemical efficiency of photosystem II, and branch water potential in the B and CONV system were similar but poor in NB treatments. Hence, this study shows that cover cropping with banding can be an efficient strategy for maintaining productivity in Fraser fir plantation.

However, there are potential drawbacks of cover crops which include heavy consumption of moisture from the soil, increased pest populations and diseases, competition with Christmas trees and reduced tree growth. White clover as ground cover decreased the height of *Abies nordmanniana* and *Abies lasiocarpa* by 30% compared to thorough weed control [43]. Both grasses and clover sown as living mulch damaged the Christmas trees more than the natural weed vegetation [43]. Christmas tree growers may incur additional costs for planting and killing cover crops.

Vegetated strips may include permanent grass strips which may take 3 to 4 years to establish and can be planted between tree rows before or after Christmas trees are planted. This permanent grass strip can significantly reduce the number of perennial and annual weeds and can also provide a stable soil surface in the winter for harvest [5].

## 7. Thermal Weed Control

Thermal weed control methods include flame, hot water, steam and infrared heater. Flaming can cause the suppression of many different annual weed species, but it is less effective on larger weeds, perennials, and grasses [54]. Flame weeding is occasionally used in Christmas tree production but needs to be applied with caution. The major challenge to flame weeding in Christmas trees is that there are many weeds at different growth stages and this method is most effective when weeds are very small (usually at the two- to three- leaf stage) [5]. This method of open flame weeding in summer can be extremely dangerous when senescing weeds and tree trimmings may provide enough fuel to ignite the entire Christmas tree field. Flame weeding in spring is more energy intensive because humidity and soil moisture are relatively high and air temperatures are low. However, an advantage of flame weeding is that the soil remains undisturbed and the chances of weed germination and emergence are reduced. Alternatives to open flame weeders are steam weeders and infrared heaters that use propane to generate heat which is applied indirectly to the weeds [5]. Solarization is another form of thermal weed control which involves heating the soil surface using plastic sheets on moist soil to trap solar radiation [55]. This method of solarization has been shown to provide effective control of many different weed species such as pigweeds (*Amaranthus* spp. L.), common purslane, and henbit (*Lamium amplexicaule* L.) [55]. However, solarization requires several weeks and warm, sunny weather to be effective [56]. Thermal weed control methods may be effective in certain cases but requires repeated applications with expensive equipment and can damage nearby Christmas trees or other materials. Hence, thermal weed control strategies may be useful in seedling or transplant production but are not common in Christmas tree production systems.

## 8. Biological Control

True biological control agents are host-specific and only attack a single species. These biological control agents contrast with generalist herbivores (or pathogens) that consume (or infect) a number

of species and are not host-specific. There is an extensive literature on potential bioherbicides, almost entirely about fungi, but there is little actual use of these as commercial or practical methods in the field [57]. Very few mycoherbicides have been registered and used commercially. DeVine [*Phytophthora palmivora* (E.J.Butler) E.J.Butler], Collego (*Colletotrichum gloeosporioides* f. sp. *aeschynomenee*), and BioMal (*Colletotrichum gloeosporioides* f. sp. *malvae*) are some examples of mycoherbicides. All of these were once registered and used commercially but subsequently withdrawn for commercial reasons [57,58]. Several insects have been used as biological agents to selectively control certain weed species. Cinnabar moth (*Tyria jacobaeae* Linnaeus, 1758) can be used to control tansy ragwort (*Senecio jacobaea* L.) [59]. The chrysomelid beetle (*Chrysolina quadrigemina* Suffrian, 1851) is used for controlling Klamath weed (*Hypericum perforatum* L.) [60]. *Tyta lactuosa* Denis and Schiffermuller, 1775, the field bindweed moth, two species of puncturevine weevils and another two species of mite can attack bindweed [5]. The authors of Reference [61] showed that the frequency of weevil (*Rhinocyllus conicus* Frölich, 1792) damage to native thistles consistently increased, reaching 16% to 77% of flowerheads per plant from 1992 to 1996. The weevils also significantly reduced seed production of native thistle flowerheads. Selection of these biological control agents is extremely important as there are instances of damage to non-target plants (Table 1).

**Table 1.** Biological control agents causing damage to non-target plants. Adapted from [62].

| Target Weed | Biological Agent | Released | Non-Targets Attacked | Damage | References |
|---|---|---|---|---|---|
| *Senecio jacobaea* L. (ragwort) | *Tyria jacobaeae* Linnaeus, 1758 (cinnabar moth) | Canada and USA, 1959–1963 | *Senecio triangularis* Hook; *Senecio integerrimus* Nutt. | not assessed or minimal | [59,63] |
| *Hypericum perforatum* L. (Klamath weed) | *Chrysolina quadrigemina* Suffrian, 1851 (chrysomelid beetle) | California, 1946 | *Hypericum calycinum* L. | marginal | [60] |
| *Carduus* spp. L. (thistles) | *Rhinocyllus conicus* Frölich, 1792 (seed weevil) | United States, 1969 | *Cirsium* spp. Mill. | not known | [64] |
| *Opuntia* spp. Mill. (prickly pears) | *Cactoblastis cactorum* Berg (internal feeding moth) | Caribbean, 1957 | *Opuntia* spp. Mill. | significant to endangered species | [65–67] |
| *Lantana camara* L. (lantana) | *Uroplata girardi* Pic. (chrysomelid leaf miner) | Hawaii, 1961, Australia, 1966 | *Ocimum basilicum* L. | minor | [68] |
| *Parthenium hysterophorus* L. (parthenium weed) | *Zygogramma bicolorata* Pallister, 1953 (chrysomelid beetle) | India, 1984 | *Helianthus annuus* L. | minor | [69,70] |

These different species of biological agents need to be released carefully and their populations should be monitored in order to achieve successful weed suppression in the Christmas tree production. Research is still continuing on many potential bioherbicides/biological control agents, but problems with mass-production, formulation, and commercialization continue to prevent their use [57,71].

## 9. Conclusions

Although there are several non-chemical methods to control weeds in Christmas tree production, proper planning and integration of different methods are required for a successful weed control program. Solely depending on non-chemical methods for controlling weeds can be labor intensive, expensive, and time consuming as constant scouting is required to reduce the weed germination and growth in the field. In most instances, integrated management is required for effective weed control. While various studies have been conducted on non-chemical methods of weed control, more research is required as it relates to Christmas tree production. Areas in which further research can improve the weed control techniques include mulch types, depths, and comparison between organic and inorganic in terms of cost, durability and control of specific weed species. More work is required on cover crop

types, competition with different species of Christmas trees and effects on seedling survival and growth. There are several potential biological agents (including both insects and fungi species), which need to be investigated more for successful weed control in Christmas tree production. Other potential future research areas include studying the effects of different irrigation practices and effects of microwave radiation to control weeds in Christmas tree production fields.

**Author Contributions:** Conceptualization, D.S.; Writing—original draft preparation, D.S.; Writing—review and editing, B.M.C., M.K.S.; Funding acquisition, D.S. All authors have read and agreed to the published version of the manuscript.

**Funding:** This work was supported by the United States Department of Agriculture (USDA) National Institute of Food and Agriculture, Hatch project number MICL02670.

**Conflicts of Interest:** The authors declare no conflict of interest.

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
