# Peer review of "A Review of Non-Chemical Weed Control Practices in Christmas Tree Production"

_forests, doi:10.3390/f11050554_

Round 1

Reviewer 1 Report

The paper covers an interesting topic, is well written and easy to read.

In some places the literature is pour but this is due to the lack of researches on the topic.

The paper need only a little adjustment.

- rows 39 and 43. As the paper will be read by international people, please add the surface in hectares in blankets.

- Figure must be cited in the main text.

- some paragraphs need to be indented (i.e. rows 153, 160, 183, 203).

Then the paper need a quick check for orthographic and punctuation error (e.g. rows 103, 142, 184, 186, 199, 295).

At last, I would ask to the authors if they have checked for researches on microwave weed control (just a curiosity, as there is a certain interest on it).

Please check the pdf file for more details.

Reviewer 2 Report

Nice paper, a good review of weed management in Christmas tree plantations. See attached pdf with comments highlighted in yellow.
